# Electric Cell-Substrate Impedance Sensing (ECIS) as a Platform for Evaluating Barrier-Function Susceptibility and Damage from Pulmonary Atelectrauma

**DOI:** 10.3390/bios12060390

**Published:** 2022-06-05

**Authors:** Eiichiro Yamaguchi, Joshua Yao, Allison Aymond, Douglas B. Chrisey, Gary F. Nieman, Jason H. T. Bates, Donald P. Gaver

**Affiliations:** 1Department of Biomedical Engineering, Tulane University, New Orleans, LA 70118, USA; jyao2@tulane.edu (J.Y.); aaymond@tulane.edu (A.A.); 2Department of Physics and Engineering Physics, Tulane University, New Orleans, LA 70118, USA; dchrisey@tulane.edu; 3Department of Surgery, Upstate Medical University, Syracuse, NY 13210, USA; niemang@upstate.edu; 4Department of Medicine, University of Vermont, Burlington, VT 05405, USA; jason.h.bates@med.uvm.edu

**Keywords:** ECIS technology, ARDS, VILI, atelectrauma, barrier function, pulmonary disease, lung, mechanical ventilation

## Abstract

Biophysical insults that either reduce barrier function (COVID-19, smoke inhalation, aspiration, and inflammation) or increase mechanical stress (surfactant dysfunction) make the lung more susceptible to atelectrauma. We investigate the susceptibility and time-dependent disruption of barrier function associated with pulmonary atelectrauma of epithelial cells that occurs in acute respiratory distress syndrome (ARDS) and ventilator-induced lung injury (VILI). This in vitro study was performed using Electric Cell-substrate Impedance Sensing (ECIS) as a noninvasive evaluating technique for repetitive stress stimulus/response on monolayers of the human lung epithelial cell line NCI-H441. Atelectrauma was mimicked through recruitment/derecruitment (RD) of a semi-infinite air bubble to the fluid-occluded micro-channel. We show that a confluent monolayer with a high level of barrier function is nearly impervious to atelectrauma for hundreds of RD events. Nevertheless, barrier function is eventually diminished, and after a critical number of RD insults, the monolayer disintegrates exponentially. Confluent layers with lower initial barrier function are less resilient. These results indicate that the first line of defense from atelectrauma resides with intercellular binding. After disruption, the epithelial layer community protection is diminished and atelectrauma ensues. ECIS may provide a platform for identifying damaging stimuli, ventilation scenarios, or pharmaceuticals that can reduce susceptibility or enhance barrier-function recovery.

## 1. Introduction

Acute respiratory distress syndrome (ARDS) is an acute form of noncardiogenic pulmonary edema that leads to flooding of the airspaces in the respiratory zone with proteinaceous fluid from the vasculature. This deactivates pulmonary surfactant, which decreases lung compliance and leads to airway closure and widespread alveolar collapse. Mechanical ventilation is necessary to sustain the critically ill ARDS patient but can cause ventilator-induced lung injury (VILI) through over-inflation of the lung tissues, leading to volutrauma, and/or repetitive airspace recruitment and derecruitment (RD), leading to atelectrauma. This contributes to an extraordinarily high ARDS mortality of nearly 40% [1].

Atelectrauma caused by (RD) is known to be injurious to the parenchyma [2,3], and is a critical contributor to the progression of VILI. Specifically, alveolar leak can result in the formation of liquid plugs in small airways, and mechanical ventilation can drive the axial movement of these plugs.

This introduces large dynamic mechanical stresses to the alveolar apical surface [4], likely due to a changing pressure gradient acting perpendicularly to the cell–surface that acts like a rolling-pin applied to the cell membrane [5]. 

While RD is clearly damaging for those with ARDS, not all RD is damaging and needs to be avoided. For example, pulmonary function tests demonstrate that closing capacity increases with age and can eventually exceed functional residual capacity, in which case airway closure occurs during normal tidal breathing. Interestingly, without concomitant damaging stimuli, this does not seem to set the aging individual up for RD damage, suggesting that the normal airway and alveolar epithelium is resilient to a certain amount and/or duration of RD. However, in other cases, RD can be quite injurious, and little is known about how resilience and damage may be a function of lung history, including exposure to deleterious chemicals or protective drugs.

Our prior studies have demonstrated through immunohistochemical visualization that RD can disrupt pulmonary epithelial barrier integrity by altering the distribution of tight-junction proteins [6]. While that work demonstrated that RD leads to barrier function disruption, the visualization technique does not allow for evaluation of the time-course of damage, which may be expanded in a vicious cycle that leads to progressive failure of the barrier [4]. This process is known as a “permeability-originated obstruction response” (POOR) that develops as a topographical growth of VILI from an initial site injured in a POOR-get-POORer mechanism [7]. In addition, little is known about how this damage may be a function of lung history, including exposure to deleterious chemicals or protective drugs.

Filling the knowledge gap related to resilience and the time-course of damage is crucial to the development of protective ventilation strategies or pharmaceuticals that prevent or delay the onset of VILI. The present study is therefore focused on three interrelated questions: Is there a method to determine the resilience of the epithelium?Under which conditions is RD likely to damage the pulmonary epithelium?What is the time course and process by which the epithelial layer becomes damaged?

Accordingly, we developed a platform that uses Electric Cell-substrate Impedance Sensing (ECIS) to quantify the onset time and subsequent rate of degradation of epithelial barrier function in an in vitro model of the pulmonary epithelium that is exposed to repetitive RD assault. 

## 2. Materials and Methods

### 2.1. In Vitro Epithelial Monolayer System

We performed our experiments in an in vitro system (Figure 1) consisting of a confluent epithelial cell monolayer cultured within a micro-channel obstructed by a liquid plug. The monolayer was comprised of NCI-H441 (HTB-174, ATCC, Manassas, VA, USA) pulmonary cells, an immortalized cell line of distal lung epithelial origin that demonstrates appreciable barrier function properties with tight-junction formation and low paracellular permeability when cultured in a confluent monolayer [8,9,10]. 

#### 2.1.1. Flow Chamber

The cells were plated into an Applied Biophysics Model 1F8 × 10E PC (Applied Biophysics, Troy, NY, USA) (Figure 2) consisting of a bottom plate with embedded electrical circuits for the impedance sensing, above which sits a Polyethylene terephthalate (PET) channel structure. Sidewalls create a uniform rectangular channel (length 50 mm × width 5.0 mm) with inlet and outlet ports. The depth of the flow chamber was set to 0.35 mm, the corresponding airway diameter in the respiratory zone of a human lung [11]. The choice of the non-air permeable PET for the channel structure was necessary in order to achieve structural rigidity to perform the RD experiment. The side-view of the flow chamber shown in Figure 2c illustrates how a finger of air penetrates the liquid-filled channel. Forward progression of the finger simulates recruitment of an obstructed airway, while retraction simulates derecruitment. 

#### 2.1.2. Preparation of the Chamber

To create a confluent monolayer, the surface of the bottom plate was pre-treated with 10 mM L-cysteine aqueous solution (Electrode-Stabilizing Solution, Applied Biophysics, Troy, NY, USA) for 20 min at room temperature. After rinsing twice with ultra-pure water, the chamber was treated with 200 µg/mL fibronectin (Plasma Human Fibronectin, Gibco, Waltham, MA, USA) in saline solution (0.15 M NaCl) for 30 min at 37 °C. The chamber was once again rinsed with the saline solution prior to seeding. The L-cystine pre-treatment conjunction with the fibronectin coat was essential to develop a confluent layer that was resilient to RD events. Preliminary RD experiments without the fibronectin treatment (data is not shown) demonstrated falloff of the cell layer at around *N* ≈ 50, regardless of the culture condition. 

#### 2.1.3. Seeding Density

In nearly all experiments, cells were cultured with an initial seeding density of 160K cell/cm^2^ suspended in cell culture medium (RPMI1640, 10% FBS). One set of experiments was conducted with a reduced seeding density of 80K cell/cm^2^ to elucidate the difference between seeding density and culture time in the creation of tight junctions. In this way, we established our protocol for a uniformly confluent monolayer by finding a minimum necessary cell seeding density to cover the channel bottom area, with the additional constraint that this layer should be developed in 24–36 h of seeding because of transport limitations inherent to the feeding condition of growth media within the airtight flow chamber. 

#### 2.1.4. Feeding Condition

To sustain adequate culture conditions within a closed chamber made of non-air permeable material, having 5 cm of length required a development of the feeding mechanism that constantly supply an oxygenated/carbonated growth media to the farthest part from the open ports_._ We tested several configurations and flow conditions to achieve a mostly uniform cell layer in the entire channel bottom surface to develop a confluent monolayer (please see the Appendix A for details). 

We attached an open reservoir on the inlet to supply well-oxygenated growth media, as shown in Figure 2c. The media was pulled from the inlet reservoir by a programmable syringe pump connected to the outlet. After an initial settling period of 10 min, a low flowrate feeding (*Q_f_* = 0.2 µL/min, average velocity *U_f_* = 1.9 µm/s) was set for 2 h, and this was followed by 2 h of *Q_f_* = 0.4 µL/min. (*U_f_* = 3.8 µm/s). After the period of initial low flowrate feeding, the flowrate was increased to *Q_f_* = 2.0 µL/min (*U_f_* = 19 µm/s) until confluence was achieved as indicated measurements of trans-epithelial impedance (see below) and visual inspection via phase contrast microscopy. 

### 2.2. Electrical Cell-Substrate Impedance Sensing

We monitored the morphological response of a cultured H441 monolayer to the RD stimulus using impedance spectroscopy with an Electrical Cell-Substrate Impedance Sensing (ECIS) device (Applied Biophysics, Troy, NY, USA), based on Giaever and Keese (1991) [12]. ECIS is a well-established label-free monitoring technique for the rapid evaluation of functional morphological changes during cell growth, wounding, and healing processes [13,14,15,16]. Since resistive and capacitive properties of different input frequencies reflect specific characteristics of inter-cellular barrier function and cell adhesion, the multiple-frequency spectroscopic measurements using ECIS are capable of evaluating stimulus/response measurements related to endothelial barrier function [17,18,19,20] and cell adhesion [21]. 

Eight electrode arrays were equally spaced along the center of the bottom plate of the micro channel, with each array consisting of ten circular electrodes (*D* = 250 µm), as shown in Figure 2b. When cells are cultured onto the bottom plate as a confluent monolayer, each electrode is covered by approximately 500–1000 cells. The cells act as insulators that modulate the impedance of a small alternating current (AC) applied across the nodes with respect to a downstream reference electrode. As illustrated schematically in Figure 3, the AC frequency affects the path of the current from the cell-covered node to the reference electrode according to the electrical properties of the monolayer. Specifically, the resistance of the monolayer at middle to low frequencies (1–2 kHz) is sensitive to the tightness of cell–cell and cell–surface interfaces, while the capacitance at high frequency (>10 kHz) is sensitive to the integrity of the cell body [22]. Multiple frequency sampling of ECIS yields the following parameters:*Rb* (Ω cm^2^) that quantifies the integrity of cell–cell tight junctions,*Cm* (µF/cm^2^) that quantifies the confluency of the cell membrane, and*α* (Ω^0.5^ cm) that quantifies cell–surface adhesion [12]. 

For real-time feedback during the experiments, we monitored cell-to-cell tight junction permeability via resistance at 1 kHz: *R*(1K) and capacitance at 64 kHz: *C*(64K). These are surrogates for *Rb* and *Cm*, respectively, which allowed us to develop a preliminary understanding of the system response. Post-processing analysis provided *Rb*, *Cm*, and *α* from recordings of impedance at 11 frequencies spanning the range 62.5 Hz to 64 kHz.

Figure 4 demonstrates that we were able to achieve a steady-state confluent cell layer by 36 h, as assessed by the convergence of the cell–layer impedance to a stable function of frequency. Specifically, by *t* = 20 h we observed the formation of an impedance plateau over the range 1 kHz < *f* < 10 kHz. This became a dominant feature at *t* = 24 h, indicative of the formation of cell–cell tight-junctions [17]. Figure 5a shows that the real part of impedance at the middle part of the frequency range is sensitive to the resistance of the intercellular path, and thus to tight-junction integrity, while Figure 5a shows that the imaginary part of impedance at high frequencies is sensitive to the capacitive behavior of the cell body, and thus to the cell coverage of the electrode.

Figure 6a,b demonstrates a similar time evolution of *R*(1K) and *C*(64K) together with a photomicrograph of a confluent monolayer at *t* = 12 h. The end of the cell attachment/spreading process was determined when *C*(64K) reached a stable flattened level. A low level of data fluctuation among nodes in *C*(64K) by *t* = 20 h indicates that the feeding strategy successfully prevented hypoxia in the farthest point from the inlet, something that can lead to regions that are sub-confluent. Confluency of the cell layer on each node was confirmed visually using phase-contrast microscopy (representative image shown in Figure 6b). It should be noted that the microscopic monitoring was only useful to detect the cell coverage (*t* < 12 h); however, the time-course of cell–cell tight junction development was determined by the behavior of *R*(1K), which reaches a sustained high level at *t >* 24 h [22,24]. Similar patterns of evolution of resistance and capacitance have been reported in other cell types [22,23,25]. 

We also tested four different cell–cell tight junction levels for *t >* 20 h based on *R*(1K) (Figure 7a) with cell layers having nearly identical confluency as demonstrated by *C*(64K) (Figure 7b). These were achieved by using seeding densities of 160k cell/cm^2^ cultured for *t* = 20, 24, and 36 h (*Case 1*, *2*, and *3,* respectively), and a reduced seeding density of 80 k cell/cm^2^ cultured for *t* = 48 h (*Case 4*). This test established that it takes at least 24 h for a uniformly fully confluent monolayer to be developed using our seeding and culture conditions, and that it takes approximately twice as long to culture a high-resistance layer when the seeding density is reduced by one-half. 

### 2.3. RD Experimental Protocol

We simulated recruitment in the cell-covered micro channel by displacing the liquid in the channel with a finger of air. This exerted mechanical stress on the monolayer (Figure 1) [26]. Retraction of the bubble then simulated derecruitment of the model airway, exerting additional stresses on the monolayer. Flow of liquid media through the channel was delivered by a computer-controlled actuator pump (LinMot; NTI AG, CH, Spreitenbach, Switzerland) for precision control of repetitive RD events via the penetration and retraction of a finger of air through the flow chamber, as illustrated in Figure 2c.

A single RD maneuver was simulated by the forward and backward motion of a finger of air that displaced the fluid from the flow chamber. The motion of the penetrating finger of air (semi-infinite bubble) was set to a constant velocity of *|U_RD_|* = 0.5 mm/s, based on physiological relevance, as discussed in [6,27]. The finger was pushed forward at a constant velocity and then retracted with an amplitude of *L* = 50 mm (full length of the flow chamber). The total RD cycle thus had a period of *T* = 20 s. 

Because the amount of air in the micro chamber influences its ECIS impedance, we alternated between brief epochs of designated RD counts designed to mechanically stimulate the monolayer and brief (~10 min) periods of multi-frequency ECIS impedance acquisition with the chamber filled with medium. This was repeated until a significant drop of *R*(1K) was observed. The number of RD events in a single epoch was adjusted to balance the operational time loss associated with impedance measurement and the accuracy of the damage classification, and typically ranged between 10–30 RD cycles/epoch.

## 3. Results

### 3.1. Experimental Data

Figure 8a,b demonstrates the behavior of a fully established confluent H441 cell layer with high levels of cell–cell binding (*t* = 36 h with 160 cell/cm^2^ seeding density) as a function of the cumulative number (*N*) of RD events. Each datapoint represents the average of three ECIS measurements of R(1K) and *C*(64K) taken after each RD epoch. Figure 8 illustrates behavior observed in all trials. Specifically, *C*(64K) was very stable for the first *N* ~ 200 cycles, indicating that the cell coverage was sustained during an initial phase of insult. Meanwhile, *R*(1K) deviated from its initial value as soon as RD occurred, but the layer sustains a high level of resistance for this initial phase, as shown in Figure 8a. After this initial phase (Phase 1), further RD stimuli caused an exponential decay of *R*(1K) to the cell-free value. Likewise, *C*(64K) increased in this secondary phase (Phase 2), indicating that cell coverage was being reduced as a result of repetitive mechanical stress. 

### 3.2. Data Analysis

From the findings illustrated in Figure 8, we hypothesized that RD initially affects intercellular binding without disrupting the confluency as measured by *C*(64K), subsequent to which cell-layer binding is diminished to a degree that allows for the cell layer to disintegrate, reducing the cell coverage. In the fully developed layer of Figure 8, Phase 1 corresponded to the first *N* ~ 200 cycles. This two-phase pattern was observed in all tests. 

Figure 9 illustrates our approach to data analysis that yielded four parameters: *R*_0_, the initial value of *R*(1K) prior to RD insults,*N_P_*_1_, the length of Phase 1, during which confluency is sustained as determined by *C*(64K),*λ*, the exponential decay constant of the monotonic reduction of *R*(1K) in Phase 2, and*R_F_*, the cell-free value of *R*(1K).

The parameter values shown in Figure 9 were computed from data from node 7 in Figure 8. The phase-contrast microscopy images shown in Figure 9 show the layers of H441 cells over the node 7 electrodes for *N* = 200 (Figure 9(1)) and *N* = 400 (Figure 9(2)). 

Phase 1 in Figure 9 corresponds to the region where *R*(1K) maintains a high and nearly constant level indicative of confluency, giving a value for *N_P_*_1_ of 208. The phase-contrast microscopic image at *N* = 200 supports the conclusion that the low level of *C*(64K) in Phase 1 represents the existence of a confluent monolayer. *R*(1K) of each node, on the other hand, varies by up to 15% and does not show a clear sign of decline until the end of Phase 1. 

Phase 2 in Figure 9 shows an exponential decay in *R*(1K) with decay constant *λ* (dimensions 1/*N*). We determined *λ* by assuming that in Phase 2 each RD insult induces a proportional reduction of *R*(1K), so that dR/dN=−λR, which provides the solution $R=R_{0}\;e^{-\lambda \left(N-N_{P1}\right)}+R_{F}$ to *R*(1K) versus *N*, where *R_F_* is the resistance of the cell-free surface, and *R*_0_
*+ R_F_* is resistance at the end of Phase 1. For the data shown in Figure 9, we obtained $R\left(1K\right)=6820\;e^{-0.0107\left(N-208\right)}+1200$ with a coefficient of determination *σ*^2^ = 0.924. The strong match of *R*(1K) to the exponential decay indicates that the monolayer in Phase 2 has very weak intercellular binding. It further suggests that the initially fully integrated cell layer at *N* = 0 becomes transformed to a layer of weak or non-existent intercellular binding with confluency at the *N* = *N_P_*_1_. Further RD insult leads to a decay of confluency where cells behave independently. The confirmation of this exponential decay strongly indicates that in Phase 2 the cells behave independently, which is consistent with a layer that lacks intercellular bonding. Figure 9-(2) clearly indicates the loss of cell coverage, which is consistent with the *C*(64K) data in Phase 2 in Figure 8b.

Figure 10 provides schematic representations of the bi-phasic RD response for cell layers that were cultured for different periods of time (20, 24, and 36 h with high seeding density) or a longer duration (48 h with a reduced seeding density), as described above. It is important to note that all layers were confluent prior to RD stimulus, based upon visual observation and *C*(64K) measurements. The key parameters demonstrated here are *N_P_*_1_ and *λ*, which are computed as described above. In this representation, we provide *R*(1K) scaled on *R*_0_. Table 1a provides the values of *N_P_*_1_ and *λ*, with the mean and standard deviation of *n* = *7* measurements. 

Table 1b provides *p* values from Student’s *t*-tests of two-tailed paired comparisons of *N_P_*_1_ and *λ*, which confirm the qualitative observations demonstrated by Figure 10 and the data in Table 1a. The *p*-values for *N_P_*_1_ indicate that even under a highly conservative criterion (*p* < 0.01), the differences in behavior between the cases are highly significant. These calculations indicate that the initial resilience of the H441 cell layer is determined by the initial state of the monolayer prior to insult. Since *C*(64K) is uniformly small and the cell layers are observed to be confluent in Phase 1, it is clear that *N_P_*_1_ is not due to a macro-scale degradation of the monolayer. In contrast, there is little or no significant difference for *λ* even for a liberal statistical significance criterion (*p* < 0.05). Evidently, *λ* displays only a weak correlation with the culture conditions—once Phase 1 is completed, the cell layers behave similarly in terms of resilience to damage from further RD insult. In Phase 2, the increase in *C*(64K) and microscopic evidence indicates that cells behave independently, and that confluence is reduced following a population-based exponential decay.

Figure 11 presents the computed values of *Rb* and *α* vs. *N/N_P_*_1_ in Phase 1. Figure 11a demonstrates that *Rb* decreases nearly linearly with *N*, indicating that the barrier function resistance created by the tight junctions is reduced by repeated RD events until the tight junction barrier function resistance is eliminated at the end of Phase 1. In contrast, Figure 11b exhibits an increase in the sub-cellular resistance with repeated RD events. 

Figure 12 demonstrates the strong correlation between *N_P_*_1_ and *Rb*_Init_, with a strikingly sigmoidal (almost step-like) response. This highly significant response (*p* << 0.001) indicates that resilience to RD injury is afforded to cell layers with *Rb*_Init_ > 5 Ω·cm^2^. Beyond this critical value, our experiments indicated that the monolayers were capable of withstanding more than 100 RD events without disassembly, while layers that do not have that level of initial barrier function resistance could tolerate only very few RD events before disassembly commences. 

## 4. Discussion

Protective ventilation strategies, such as positive airway pressure (PAP), positive end-expiratory pressure (PEEP), and airway pressure release ventilation (APRV) are all designed to reduce atelectrauma by keeping the airway pressure high enough to prevent re-closure during the exhalation phase [28,29]. Nevertheless, mortality rates have not decreased with these protective strategies, perhaps because of their inability to control events of disparate timescales in a heterogeneous environment [30,31]. Accordingly, to gain a deeper understanding of the damaging effects of RD, we cultured monolayers of H441 cells that exhibited an appropriate level of integrity against which mechanical stress could be tested [32]. This was a non-trivial undertaking because of the low surface area/volume ratio within the ECIS flow chamber, and the fact that the rigidity of the top structure provided by PET is essential to ensure that the controlled air–liquid interface motions did not mechanically distort the structure. For this reason, the adaptation of the chamber to incorporate an air-permeable polymer, such as poly-dimethyl-siloxane (PDMS), for the construction of the top plate of the chamber, was ruled out due to its low modulus. 

Our preliminary studies demonstrated the critical importance of developing a uniformly consistent monolayer of epithelial cells within the flow chamber, as exists in pulmonary airways and alveoli. NCI-H441 epithelial cells are known to form ‘multiple layers’ if an excessive cell density is introduced at beginning of the inoculation process, showing that morphologic, phenotypic, and functional characteristics are strongly influenced by culture condition [10]. Multi-layers are non-physiological and are also not as susceptible to barrier function disruption compared to monolayers, so we took great care in developing uniformly confluent monolayers. 

By comparing *N_P_*_1_ with *R*_0_ (the initial value of *R*(1K) prior to insult, Figure 7a), it is evident that a qualitative correlation exists between *N_P_*_1_ and *R*_0_. Although statistical significance is lacking because of the large standard deviations in *R*_0_, this observation suggests that the initial resilience is determined by how long intercellular binding can hold the layer together. Once this intercellular binding is disassembled, the layer becomes highly vulnerable to cell membrane damage or cell removal, as demonstrated by *C*(64K). 

To test the hypothesis that intercellular binding determines the cell-layer resilience to RD, we utilized the post-processed multi-frequency ECIS data to separate the capacitance and electrical resistances in the sub- and intercellular pathways using the Giaever-Keese model [12], which is based on a transfer function set by a cell layer modeled as circular disks of radius *r_c_* elevated by a distance *h* above the surface of the electrode. The value of the membrane capacitance, *Cm*, is a measure of the capacitive characteristic influenced by composition and morphology of the apical membrane surface [17]. Since the current study used only one type of cell line, a cell membrane capacitance value (*Cm*) did not create a meaningful difference in Phase 1 once the layer achieved confluency, and was consistent with typical *Cm* values [33]. *Cm* data (not shown) were nearly identical to the behavior of *C*(64K) in Phase 1, indicating that cell confluency was sustained during the initial stages of RD. The two resistance parameters extracted from this analysis are *Rb*, a measure of the electrical resistance in the intercellular pathway reflective of tight junction integrity, and *α*, a measure of electrical resistance in sub-cellular space of depth *h*. Both *α* and *Rb* contribute to *R*(1K) (Figure 3).

Following the Giaever–Keese model, $\alpha =r_{c}\sqrt{\rho /h}$, where *r_c_* is the cell radius, *ρ* is the specific resistivity, and *h* is the substrate distance between the electrode surface and the cell body [17]. It is likely that the compression of the monolayer from the rolling-pin-like pressure field surrounding the finger of air leads to a compression of *h.* This occurs with each RD event, because a normal stress $\tau_{n}$ is exerted on the cell layer that is proportional to the Laplace pressure drop $\Delta P\propto \gamma /r_{fc}$ (Figure 1). This pressure squeezes liquid from the basal side of the cell-layer, decreasing *h* and commensurately increasing *α.* Although we cannot completely rule out the possibility of cell membrane damage in Phase 1 or a remaining impact of the intercellular binding in Phase 2, our data demonstrate that the repetitive input of mechanical stress causes degradation of the cell–cell tight junctions prior to the induction of cell-layer disintegration. In the other words, a confluent epithelial cell layer with a damaged tight junction is highly vulnerable to airway reopening and closure events that occur during atelectrauma. 

The duration of Phase 1 (*N_P_*_1_) is the parameter most reflective of resilience to the repetitive RD stress stimuli since, during this phase, community interactions exist that protect the cell layer from disassembly. Furthermore, we hypothesize that the value of *Rb* prior to RD, *Rb*_Init_, determines *N_P_*_1_. Accordingly, we calculated *Rb*_Init_ for the experimental trials with the different culture conditions described above, as shown in Figure 12. The data presented in Figure 11 demonstrate why *R*(1K) is insensitive to changes in the barrier function resistance, *Rb*. Specifically, since R(1K)=f(Rb, α), the reduction of *Rb* and simultaneous increase in *α* reduces the sensitivity of *R*(1K) to modifications of intercellular binding interactions that are impacted by RD events. 

The big picture of Phase 1, therefore, is that there is a consistent and nearly linear reduction of *Rb* with repeated RD. This indicates that the barrier function resistance associated with tight-junction proteins is systematically reduced until the tight junctions that suture the cell layer together as a community are eliminated. This leads to the Phase 2 response in which cells behave independently, which initiates the exponentially decreasing *R*(1K) and increasing *C*(64K), as cells are systematically eliminated from the monolayer during Phase 2 RD events. In addition, the observed changes of *Rb* and *α*, during the early stage of Phase 1 (*N/N_P_*_1_ < 0.1), may indicate how trans-membrane proteins or peripheral membrane proteins respond to the imposed dynamic pressure gradient on the cell layer. The time scale of the response (~minutes) is consistent with observations of cellular adaptation to mechanical stress [34,35] rather than wound healing or migration mechanisms of injured lung epithelium (hours ~ days) [36,37]. 

The stimulus/response curve shown in Figure 12 represents a new technique for identifying the method and processes by which recruitment and derecruitment induce atelectrauma, which is a fundamental cause of ventilator-induced lung injury. In particular, this new technique may provide a tool for assessing how tolerable atelectasis is in an ARDS lung, and how likely it is that damage will result from repeated RD events. This is critical to avoiding the POOR-getting-POORer scenario that can lead to the lung entering into the ‘VILI vortex’ that is so often fatal [4,7,28,38]. 

Nevertheless, the present study has some limitations. For example, we only studied a single simulated airway geometry, but the caliber of the airway/alveolus (*R*) determines the magnitude of the Laplace pressure drop, $\Delta P=\frac{\gamma }{R},$ where *γ* is the surface tension [5,27,39,40]. The stress stimulus also depends on the dimensionless velocity associated with RD, $Ca=\frac{\mu U}{\gamma }$. *Ca* is known as the ‘capillary number’, which is a function of the lining fluid viscosity (*μ*), the RD velocity (*U*), and the surface tension (*γ*). We also did not investigate the extent to which a decrease in *Ca* is associated with increased cellular damage, which is to be expected due to the creation of a large rolling-pin-like pressure gradient that sweeps across the lumenal surface. In addition, we did not examine the effects of changing *γ*, which is a dynamic variable that depends upon the rate and history of the breathing cycle. In the healthy lung, surface tension varies during the breathing cycle to stabilize the lung; however, in ARDS the surface tension is increased, and the dynamic variation is stunted in a manner that leads to atelectasis and induces an increase in the magnitude of mechanical stress [41]. Indeed, the stimulus/response behavior may become altered as a function of the biological properties of the monolayer and flexibility of the substrate. 

We did not conduct long-term monitoring and observation of the epithelial layer with different degrees of damage. Repair of tight-junctions could be observable in Phase 1 as a post-processed evaluation of Rb. In contrast, Phase 2 in Figure 9 illustrates layer dis-integration; therefore, recovery from this state would require cell division and/or migration to the electrode, a process that takes days.

In the present study, we explored the resilience of a confluent monolayer of immortalized H441 cells, but this cell line is unrealistically uniform; the pulmonary epithelial surface consists of a diverse population of cells that may be either more or less resilient than the uniform layer investigated herein. Concomitant pathophysiological processes such as inflammation may significantly increase the susceptibility of the lung to injury from RD events that, when coupled with the mechanical stresses associated with atelectrauma that likely deform and strain the epithelial cells and substrate [42], will lead to biological signaling that could either reinforce or damage intercellular binding. We can thus expect that the stimulus/response behavior of the real airway epithelium will be less binary than that observed in the present study. 

## 5. Conclusions

In conclusion, we have shown that maintenance of robust intercellular binding is a critical determinant of epithelial monolayer resilience in the face of damaging mechanical stresses associated with atelectrauma, and that RD-induced damage to the lung epithelium progresses in two phases. During the initial Phase 1, confluence is maintained but measurements of intercellular binding resistance indicate that damage to the integrity of the epithelial barrier is declining. During Phase 2, monolayer confluence decreases progressively, indicating that intercellular binding becomes weak or even nonexistent. These findings are consistent with clinical evidence of the development of atelectrauma in the ARDS lung, demonstrating that atelectrauma does not develop as soon as injurious mechanical ventilation is instigated but rather is initiated after a period of time in collapsed respiratory zones where the permeability of the epithelial cell layer is already low [43]. Importantly, biophysical insults that either reduce barrier function or increase mechanical stress may make the lung more susceptible to atelectrauma. 

ECIS may provide a platform for identifying the RD-associated stimuli that damage the lung epithelium. Expansion of the current method to drug testing will allow us to monitor the time-course of the stimulus/response behavior to complex lung injury processes, in vitro. Information regarding the time-course and pattern of damage, and not just a degree of damage, could be highly valuable. For example, this process could be adapted for the use of rapid in vitro screening to develop ventilation scenarios or pharmaceuticals that might reduce either the susceptibility to damage, or enhance barrier-function recovery. Effective development of this ECIS platform could therefore elucidate methods to reduce the duration and/or severity of ARDS and VILI.

## Figures and Tables

**Figure 1 biosensors-12-00390-f001:**
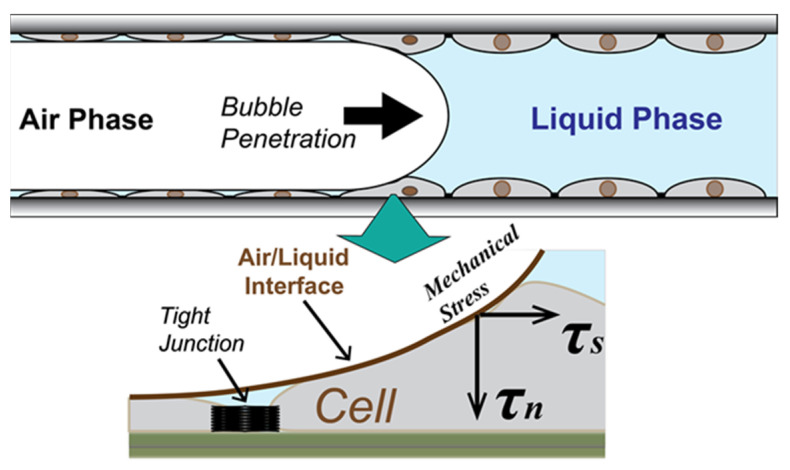
Schematic of stress on epithelial cell layer in the atelectrauma zone (not to scale). The recruitment and de-recruitment of air/liquid interface generate repetitive input of the mechanical stress on a membrane and tight junction of the wounded epithelial cell layer.

**Figure 2 biosensors-12-00390-f002:**
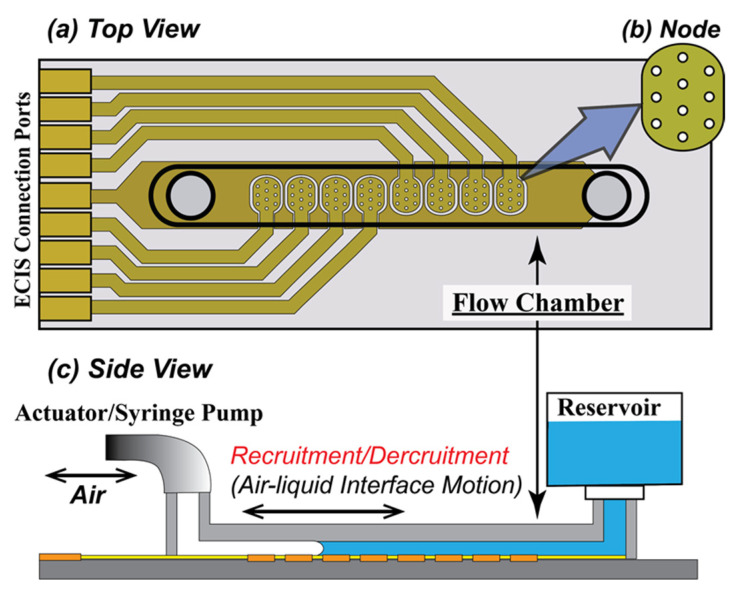
Schematic of (**a**) the ECIS Flow Device, (**b**) Node with 10 electrodes, and (**c**) setup of the cell culture system and the RD experiment. (**a**) The ECIS flow device has a single straight channel with length × width × depth = 50 mm × 5 mm × 0.36 mm. (**b**) Node 1 to 8 (left to right) are placed along the axial direction, covering approximately 70% of the bottom where the H441 cell layer is cultured. Each node has 10 electrodes, having a 250 µm diameter. (**c**) Growth media fills the channel from the reservoir during the culturing of a confluent monolayer that covers the electrodes. Recruitment is induced by a penetrating finger of air that is driven by a computer-controlled actuator pump. A reverse motion reintroduces liquid to complete one RD cycle. The device is tilted at a 45° angle to avoid bubble break-up during the RD motion.

**Figure 3 biosensors-12-00390-f003:**
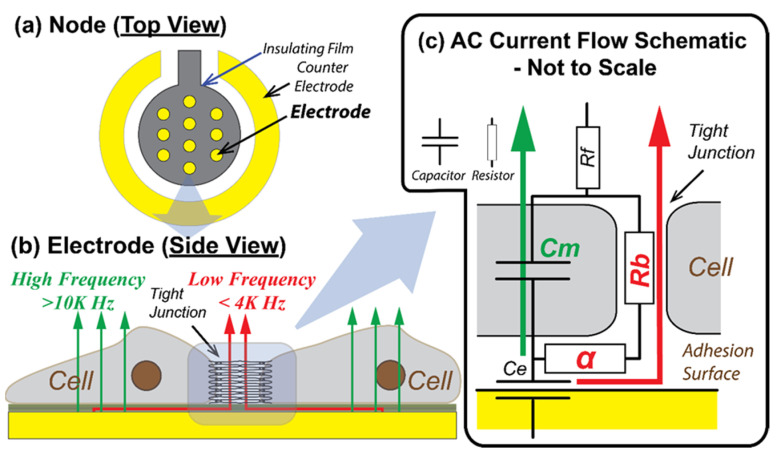
Schematics of ECIS electrode configuration and AC current flow paths (not to scale, based on [12,17]). (**a**) The node consists of 10 electrodes, each of which has a diameter of 250 µm providing the surface area to be covered with 100–1000 cells. (**b**) At relatively low frequencies (<2 kHz) most of the current flows under and between adjacent cells (through tight junctions), and at higher frequencies (>20 kHz) more current passes directly through the insulating cell membranes [23]. (**c**) An explanatory simplified electrical circuit to demonstrate the impedance modeling parameters in the current flows through and around cells. The modeling parameters of basal adhesion function (*α*) and intercellular barrier function (*Rb*) are represented as resistors, since both parameters are dependent on the binding and the electrical pathway gap. Cell membrane capacitance (*Cm*) is dependent on the composition of the cell membrane. *Ce* is the electrode capacitance and *Rf* is the resistance of the growth media. In the current study, we use resistance at frequency 1 kHz, *R*(1K), to monitor the formation of cell–cell tight junction, and capacitance at 64 kHz, *C*(64K), to monitor the formation of confluent cell layer on electrodes.

**Figure 4 biosensors-12-00390-f004:**
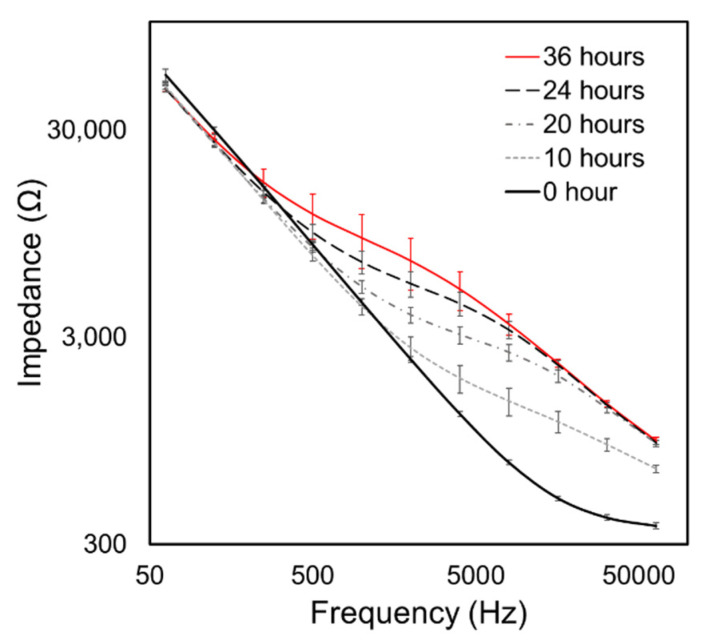
The evolution of Impedance vs. Frequency throughout the culturing period of 36 h. Error bars are standard deviations of data from *n* = 8 electrode arrays in the flow chamber. Our results are similar to evolution patterns by [17]. The initial spreading period occurs over *t* < 10 h, leading to the impedance increasing over 10 kHz < *f* < 64 kHz. Subsequently (*t* > 20 h), the impedance of the middle frequency region (1 kHz < *f* < 10 kHz) elevates, which represents formation of cell–cell binding.

**Figure 5 biosensors-12-00390-f005:**
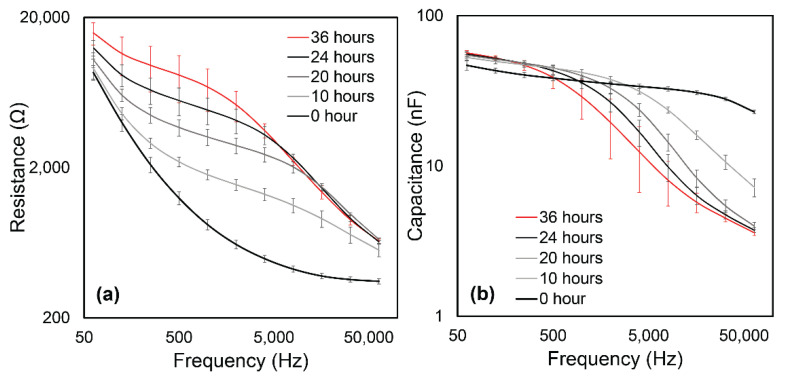
(**a**) Resistance and (**b**) Capacitance evolution curves derived from Figure 4. (**a**) The frequency response of resistance in the middle to low range (1 kHz < *f* < 10 kHz) is sensitive to the electrical pathway between cells and the channel surface. This evolution demonstrates the creation of cell–cell binding. (**b**) The capacitance at high frequency current (*f* > 10 kHz) is sensitive to the current passing through the cell body and quantifies confluence.

**Figure 6 biosensors-12-00390-f006:**
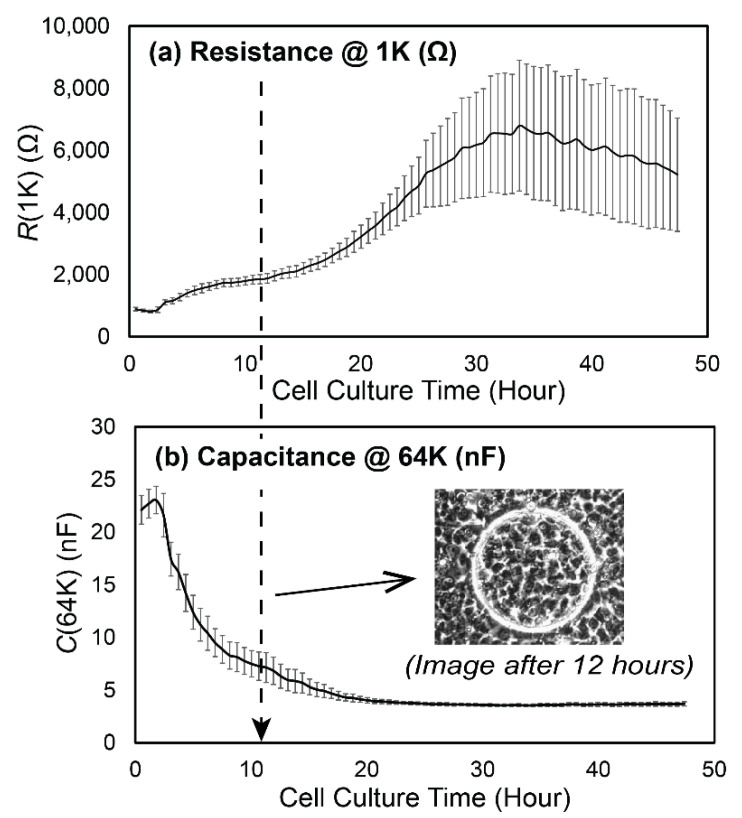
Time evolution of (**a**) *R*(1K) and (**b**) *C*(64K) as the cell layer was cultured. We use *R*(1K) to monitor the integrity of cell–cell tight junction, and *C*(64K) to monitor the cell layer confluence. The microscopic image in (**b**) is a representative image of the layer confluency at *t* = 12 h. Error bars are standard deviations of the *n* = 8 nodes in the flow chamber.

**Figure 7 biosensors-12-00390-f007:**
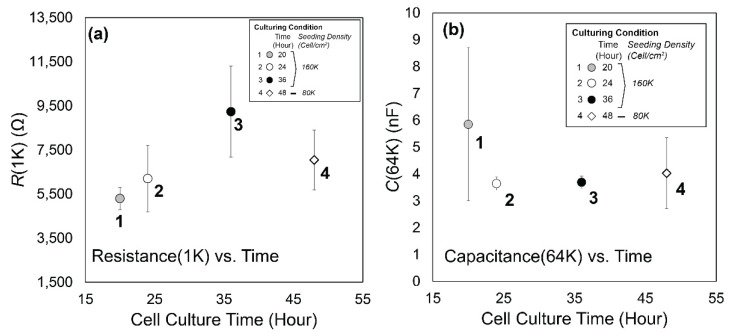
(**a**) *R*(1K) and (**b**) *C*(64K) comparisons of four H441 confluent layers having different cell–cell tight junction density generated by using different culture times and seeding densities. Error bars represent standard deviation of data from the nodes (*n* = 7 or 8) in the flow chamber.

**Figure 8 biosensors-12-00390-f008:**
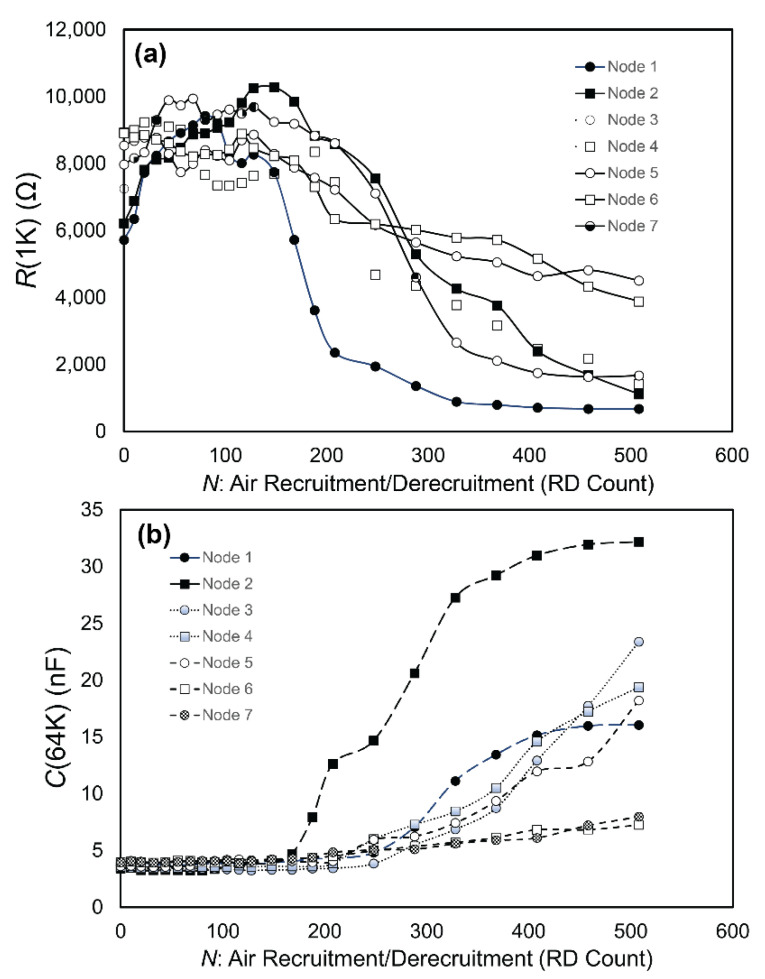
Representative measurements of RD damage progressions on (**a**) *R*(1K) and (**b**) *C*(64K). *C*(64K) suggests that the cell layer coverage does not show signs of reduction until the RD cycle 170 < *N* < 250, which determines the length of Phase 1, *N_P_*_1_. Within Phase 1, *R*(1K) varies by ~15%. For *N* > *N_P_*_1_, *R*(1K) and *C*(64K) deviate exponentially from Phase 1 values, indicating a loss of confluency of the cell layer.

**Figure 9 biosensors-12-00390-f009:**
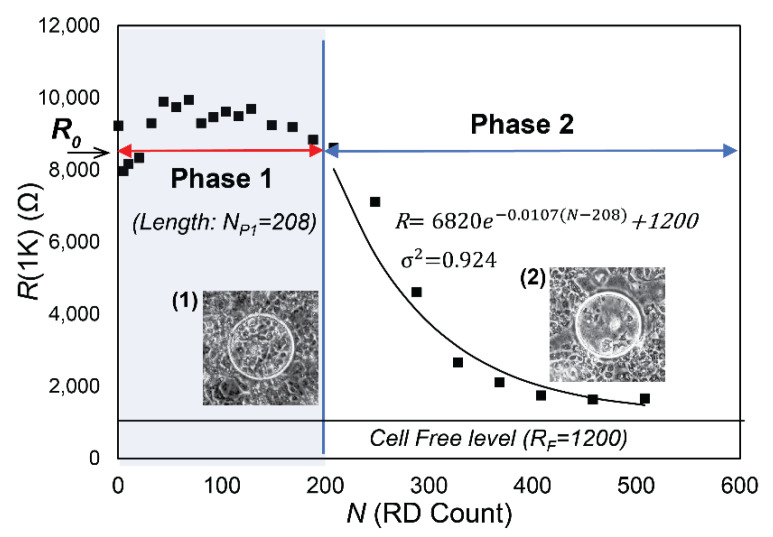
Schematic representation of *N_P_*_1_ and *λ* with *R*(1K) plot. Phase 1 is characterized by a consistent *R*(1K) value (with variations of 10–15%). The microscopic observation (1) demonstrates confluency on the circular electrode. Phase 2 demonstrates an exponential decay to the cell free value (*R_F_*).

**Figure 10 biosensors-12-00390-f010:**
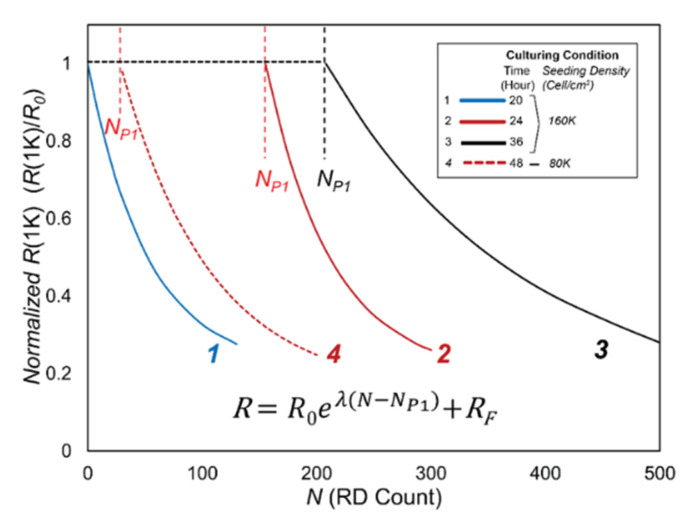
Schematic representation of Phase 1 and Phase 2 behavior based on functions in Table 1.

**Figure 11 biosensors-12-00390-f011:**
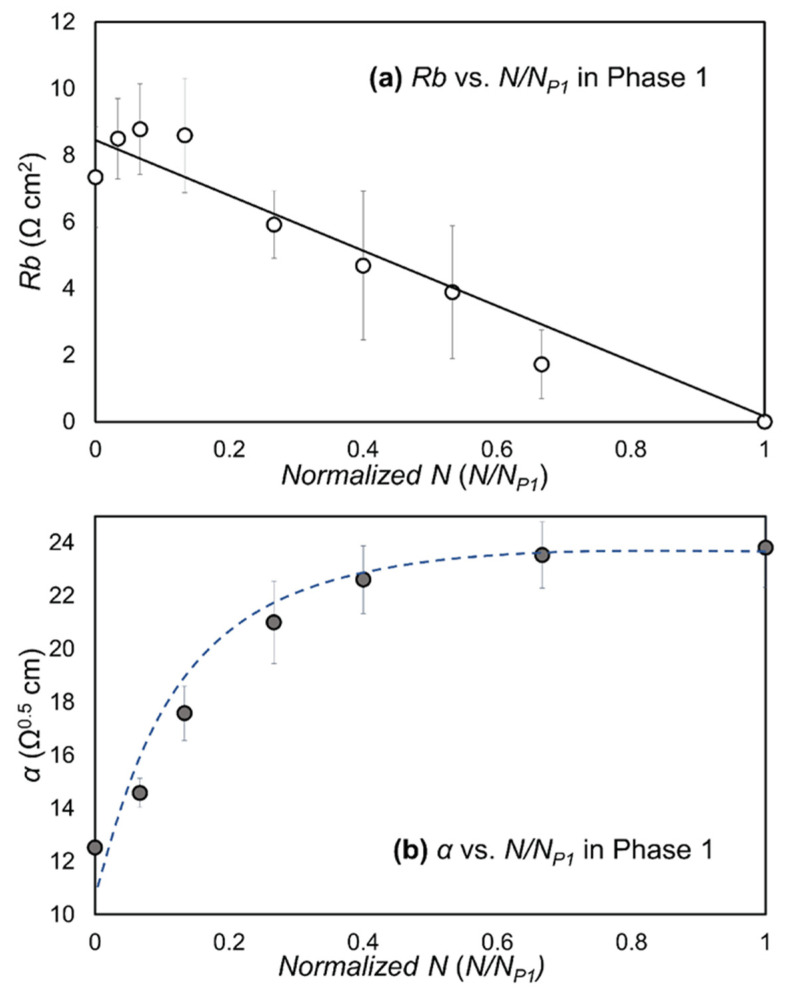
Plots of (**a**) *Rb* vs. *N/N_P_*_1_ and (**b**) *α* vs. *N/N_P_*_1_ in Phase 1. Error bars are standard deviations between nodes. Trendline for *Rb* is based on a linear regression.

**Figure 12 biosensors-12-00390-f012:**
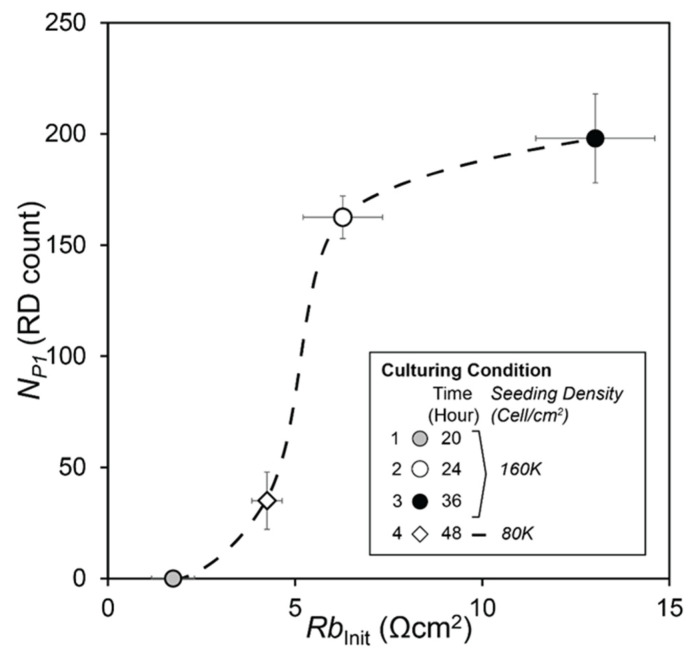
*N_P_*_1_ vs. *Rb_Init_.* The Phase 1 length, N_P1_, has strong correlation with *Rb_Ini_*_t_, with *p* << 0.001.

**Table 1 biosensors-12-00390-t001:** (**a**) Computed *N_P_*_1_ and *λ.* S. Density is the seeding density, C. Time is the culture time in the flow chamber, and std is standard deviation. (**b**) *p*-values of corresponding case.

(a)	Case	S.Density	C.Time	*N_P1_*		*λ*		
		(Cell/cm^2^)	(hour)	(N)	*std*	(N ^−1^)	*std*	
	**1**	160K	20	0	0	−0.0192	0.0154	
	**2**	160K	24	143	9.6	−0.0173	0.0120	
	**3**	160K	36	198	20	−0.0059	0.0051	
	**4**	80K	48	35	12.9	−0.0127	0.0054	
(**b**)	** *N_P1_* **				** *λ* **			
	**Case**	1	**2**	**3**	**Case**	**1**	**2**	**3**
	**2**	1.66 × 10^−5^			**2**	1.42 × 10^−1^		
	**3**	7.78 × 10^−6^	5.62 × 10^−3^		**3**	1.95 × 10^−1^	2.44 × 10^−2^	
	**4**	6.18 × 10^−4^	1.15× 10^−4^	1.18 × 10^−4^	**4**	7.88 × 10^−2^	1.31 × 10^−2^	4.40 × 10^−1^

## Data Availability

Original data from this study is held by the lead author Eiichiro Yamaguchi and can be obtained by request at guchi@tulane.edu.

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
