# Peer review of "Electric Cell-Substrate Impedance Sensing (ECIS) as a Platform for Evaluating Barrier-Function Susceptibility and Damage from Pulmonary Atelectrauma"

_biosensors, 2022, doi:10.3390/bios12060390_

Round 1

Reviewer 1 Report

This manuscript described the using of ECIS to evaluate barrier-function susceptibility and damage from pulmonary atelectrauma. The topic is inetersting. Soem results have been obtaied. However, some points needs to be addressed:

1. In Fig. 7a, the abscissa axis is covered by values, and the group diagram is improper. 
2. Good monolayer cells must be formed within 36h. After 24h, the tightly connected monolayer cells reach the saturated state, and after 12h, the cell adhesion reaches the stable stage. This paragraph shows that in addition to the measurement of the cell resistance, it is more convincing to have the corresponding microscope diagram. There are some obvious bubbles in the microscope diagram as shown in Figure 6. After magnification, only the cell connection can be seen, but the cell state can not be observed. If the cell state is poor, the corresponding connection will be weakened, and the three-dimensional structure and permeability will change.
3. The finger of air is put into the simulated RD to detect cell damage. However, as the author mentioned above, the tracheoalveolar epithelium may damage and repair itself for a certain amount and / or duration of RD. Whether it is necessary to set the number of RD cycles after which the epithelial cells can no longer recover themselves. The manuscript does not set up long-term monitoring and observation of cells with different degrees of damage.
4. Does the integrity of cell bodies or other factors in Phase2 lead to the weakening of intercellular binding?
5. It is better to provide  high-resolution physical map and cell attachment pictures of one node observed under the microscope. 
6. line136-139, the basis for using 0.2 µl/min, 0.4 µl/min and 2.0 µl/min of Qf should be described and related references are needed. 
7. line 151: ......ECIS have been used to evaluate stimulus/response......

Reviewer 2 Report

The paper reports a very interesting in vitro study about ECIS as a experimental technique to quantify lung damage during and after forced ventilation. The work is well written, methods and procedure are described in details. In my opinion, the article can be published. I only suggest the authors to better explain and comment about how the ECIS can be applied in clinical routine in order to give quantitative indication on patients subjected to forced ventilation.

Round 2

Reviewer 1 Report

Tha authors have addressed most of my questions.